Analysis of factors contributing to postoperative body weight change in patients with gastric cancer: based on generalized estimation equation

Tian Qiuju 1
Qin Liyuan 1
Zhu Weiyi 2
Xiong Shaojie 3
Wu Beiwen wbw20578@rjh.com.cn gaoan2005new@163.com 2
1 Nursing college, Shanghai Jiao Tong University School of Medicine , Shanghai , China
2 Nursing Department, Shanghai Jiao Tong University School of Medicine, Ruijin Hospital , Shanghai , China
3 Gastrointestinal surgery department, Shanghai Jiao Tong University School of Medicine, Ruijin Hospital , Shanghai , China
Keogh Justin
Electronic publication date: 2020 Jul 10
Publication date: 2020
Volume: 8
Electronic Location ID: e9390
Received 2019 Oct 30; Accepted 2020 May 28
Copyright: ©2020 Tian et al.
Copyright year: 2020
Copyright holder: Tian et al.
License: This is an open access article distributed under the terms of the Creative Commons Attribution License, which permits unrestricted use, distribution, reproduction and adaptation in any medium and for any purpose provided that it is properly attributed. For attribution, the original author(s), title, publication source (PeerJ) and either DOI or URL of the article must be cited.
License URL: https://creativecommons.org/licenses/by/4.0/

Keywords: Gastric cancer, Body weight loss, Gastrectomy, Body mass index

Funding: Shanghai Municipal Commission of Health and Family Planning, Key developing disciplines 2015ZB0305 This work was supported by the Shanghai Municipal Commission of Health and Family Planning, Key developing disciplines (NO: 2015ZB0305). The funders had no role in study design, data collection and analysis, decision to publish, or preparation of the manuscript.

==============================
Aims

The study aimed to explore factors contributing to body weight change over time in gastric cancer patients after gastrectomy, in order to find risk factors to implement nutritional intervention beforehand.

Methods

A cohort of gastric cancer patients who were treated with gastrectomy from January to March 2019 at a university affiliated hospital in Shanghai were consecutively identified in this study. Demographics, disease related information, nutrition knowledge, attitude, and practice score were collected before gastrectomy. In addition, body weight before surgery (T0), body weight at one month (T1), two months (T2), and three months (T3) after gastrectomy were recorded. Generalized estimation equation was used to describe body weight change and analyze factors contributing to body weight change after surgery.

Results

There were 49 patients recruited in the study. Patient body weight decreased by 9.2% at T1 (Wald χ = 271.173, P <0.001), 11.0% at T2 (Wald χ2 = 277.267, P <0.001), and 11.4% at T3 compared to baseline at T0 (Wald χ = 284.076, P <0.001). The results of GEE for multivariable analysis showed that surgery type (Wald χ = 6.027, P = 0.014) and preoperative BMI (Wald χ = 12.662, P = 0.005) were contributing factors of body weight change. Compared with distal gastrectomy patients, total gastrectomy patients experienced greater body weight loss (β = 2.8%, P = 0.014). Compared with patients with BMI&λτ; 18.5 kg/m2, patients with BMI ≥ 25 kg/m2experienced greater body weight loss (β = 4.5% P = 0.026).

Conclusion

Gastric cancer patients experienced significant weight loss during 3 months after gastrectomy. Total gastrectomy and BMI ≥ 25 kg/m2were risk factors to postoperative body weight loss for GC patients. The results suggested hinted that clinician should pay attention to postoperative nutrition status of patient undergoing total gastrectomy and obesity patients.

Introduction

Although the morbidity and mortality of gastric cancer (GC) have been decreased recent years globally, new cases and deaths from GC in China are still on a relatively high level, accounting for 42.6% new cases and 45% deaths from GC worldwide (Ferlay et al., 2015). As the development of medical science, GC patients have experienced longer survival (Allemani et al., 2018). In China, five-year survival rate for GC increased from 27.4% in 2003 to 35.1% in 2015 (Zeng et al., 2018).

Surgical resection is considered to be the only radical treatment for GC combined with neoadjuvant therapy or not. Body weight loss is a prevalent problem among patients after surgical treatment. A study in Japan followed up 105 patients with stage I GC undergoing open distal gastrectomy (ODG) or laparoscopy-assisted distal gastrectomy (LADG), and found that body weight loss at 1 week, 1 month, and 3 months were –3.0%, –4.9%, and –5.4%, respectively in patients who underwent ODG, while in those undergoing LADG, the body weight change was –2.7%, –4.3%, and –5.7%, respectively (Aoyama et al., 2018). Patients who were treated with neoadjuvant chemotherapy before gastrectomy experienced a nadir weight loss about 10%–15% at 6 months after surgery (Davis et al., 2016). In China, Du et al. (2019) found that body weight loss of GC patients reached the lowest within 3 months after gastrectomy.

Body weight loss has been associated with GC patient outcome. A more than 12% post-surgery body weight loss was strongly associated with poorer disease-free survival in GC (Kubo et al., 2016). In addition, body weight loss has not only been related with GC patient postoperative survival, but also associated with poorer quality of life. A cohort study followed up 76 patients undergoing curative gastric cancer resection and surviving at least two years without recurrence to examine the quality of life, and the results showed that body weight loss 10% or greater was associated with deterioration of all functional aspects of quality of life, as well as persistent pain, diarrhea, and nausea/vomiting (Climent et al., 2017). Furthermore, body weight loss reduce tolerance to treatment. Aoyama et al. found that body weight loss after gastrectomy in GC patients was an independent risk factor for the continuation of S-1 adjuvant chemotherapy (Aoyama et al., 2013). Therefore, considering the high prevalence of weight loss and its negative effect on outcome, it is necessary to identify the factors contributing to postoperative body weight change to improve the outcome and quality of life of the patients.

Body weight change in GC patents after gastrectomy involves tumor-, treatment-, and nutrition- related factors. Identifying and predicting patients at risk of postoperative body weight loss are limited. Davis et al. (2016) analyzed factors contributing to postoperative weight loss, and found that the extend of weight loss after gastric cancer at one year depended on preoperative body mass index (BMI) and extent of gastric resection. Segami et al., (2018) detected that grade 2 or higher post-operative complication according to Clavien-Dindo classification and total gastrectomy were significant risk factors for severe body weight loss (more than 10% body weight loss) during the first month after surgery. Tanabe et al. (2017) found that higher preoperative BMI, total gastrectomy, and female sex were independent predictors of greater body weight loss at the first one year mark after gastrectomy. These studies have focused on the demographics, clinicopathology as well as treatment related factors on body weight loss, but some findings of these studies were inconsistent. Besides, patient’s nutrition knowledge, attitude, and practice as internal impetus, were an important part of nutrition therapy. Patients with higher nutrition literacy would take active actions in dietary intake, which may result in body weight change. Furthermore, most of the studies on the factors affecting patient’s body weight after gastrectomy were based on the measured body weight value at a certain time or the difference between the two measured values, and the analysis was inclined to a fixed time point. There is currently no multi time-point measurement to analyze the factors contributing to body weight change at multiple time points after gastrectomy in GC patients.

Hence, this study aimed to prospectively explore the influence of preoperative BMI, patient’s knowledge, brief, practice score, demographics, clinicopathology, and treatment related factors on body weight change during three months after gastrectomy in patients with GC.

Material and Methods

Study design

Prospective longitudinal experimental design was used to identify factors contributing to body weight change during three months after gastrectomy in GC patients. Ethics approval of this study was obtained from Ruijin Hospital Ethics Committee, Shanghai Jiao Tong University School of Medicine (Shanghai, China).

Participant recruitment

GC Patients undergoing gastrectomy between January to March 2019 were identified in gastrointestinal surgery department prospectively. Volunteers meeting the following criteria were recruited: (1) diagnosed with GC; (2) scheduled for open gastrectomy; (3) aged 18 or above; (4) informed and agreed to participate in this study. Participates were excluded when they met condition below: (1) combined with other malignancies besides GC except distant metastasis; (2) palliative treatment; (3) recurrence within residual stomach; (4) patients with tube feeding after discharge. A written informed consent was obtained from all patients enrolled in the study. The sample size was estimated by G*power software (version 3.1) with three repeated measures, within factors design, a power of 0.80, significant level of 0.05, and effect size of 0.30.A sample size of 43 was required for analyzing body weight change during three months after gastrectomy by considering the attrition rate of 10%.

Data collection

Volunteers were asked to complete a demographic questionnaire and a Digestive Cancer Patients Nutrition Knowledge, Attitude and Practice Questionnaire (DCNKAPQ). DCNKAPQ was designed to assess digestive cancer patients’ nutrition knowledge, nutrition attitude, and nutrition practice with Cronbach’s alpha 0.822 (Jing, Wei-li & Xin-qiong, 2016). Specifically, nutrition knowledge dimension contains 17 questions, one point would be taken when the question was given the right answer, otherwise 0. Nutrition attitude dimension includes 5 items, which are scored on a 0-4 response scale, with answers ranging from “strongly disagree” to “strongly agree”. Nutrition practice dimension covers 8 items, which are scored on a 0-4 response scale, with answers ranging from “never” to “always”. Total scores of DCNKAPQ range from 0 to 69, with score 0-17 from nutrition knowledge, score 0-20 from nutrition attitude, and 0-32 from nutrition practice. The demographic information included age, height, body weight, gender, marriage status, educational level, and religious faith.

Data related with disease and treatment variables were collected from electronic patient record, including cancer stage, cancer treatment, postoperative complication, nutrition support at home, and comorbidity. The tumor staging was evaluated using the 7th edition tumor-node-metastasis (TNM). The Charlson comorbidity index (CCI) was applied to represent patient’s comorbidity severity. Categories of preoperative BMI were based on Asian standards proposed by WHO obesity expert advisory group in 2002, where BMI < 18.5 kg/m2, BMI between 18.5 kg/m2 to 22.99 kg/m2, BMI between 23 kg/m2 to 24.99 kg/m2, and BMI ≥ 25 kg/m2 indicate underweight, normal weight, overweight, and obesity, respectively. Body weight were collected at admission (T0), one month (T1), two month (T2), and three month (T3) after surgery. The body weight at four time points were reported by the patient self. The patients were required to wear the same outfit without accessories every time they measured the body weight for technical consistency. The percentage of body weight loss at each point was defined as follows: (preoperative body weight –body weight at each point after surgery) *100% / preoperative body weight.

Statistical analysis

Data analyses were performed by IBM SPSS 20.0. Measurement data meeting normal distribution were expressed as mean ± standard deviation (SD), otherwise median combined with 25% quartiles and 75% quartiles. Categorical data were showed in the form of frequency or percentage. Generalized estimating equations (GEEs) were used to analyze the univariate and multivariable influence of demographic variables and clinical variables on body weight change in GC patients over time. Variables with at least a significant of P value less than 0.1 on univariate analysis (all characteristic, disease-, treatment-, nutritional variables) went on to be included in multivariate analysis. Quasi-likelihood under the independence model criterion (QIC) value is an index to evaluate whether working correlation matrix is appropriate for GEE. The lower the QIC value is, the fitter the GEE parameter estimation is. All statistics were based on two-sided test with level of significance set at 0.05.

Results

Baseline characteristics of the study population

Eighty-two potentially eligible patients were available, but 27 patients did not meet the eligibility criteria, and 6 patients did not participate in follow up from T1 to T3. A total of 49 patients were involved in the study, including 1 patient lost at T1, but recovered at T2 and T3. There was no case of relapse observed during the follow-up period. The median age of them was 57 years old (range 29-75 years old), the median preoperative body weight of male was 69.25 (62.50, 72.50) kg in male GC patients and 57.00 (51.00, 61.00) kg in female GC patients. The majority of them were married (91.80%), without specific religious faith (85.71%), and with a stage III or IV cancer (95.92%). There were 13 cases with postoperative complications, including 6 cases with infection, 3 cases with delayed gastric empty, 1 case with diarrhea, 1 with bleeding, 1 with duodenal stump fistula, and 1 with deep vein thrombosis. The median value of CCI, nutrition knowledge, attitude, and practice score were 5.00 (3.00, 6.00) points, 12.00 (8.00, 13.00) points,15.00 (14.00, 18.00) points, and 21.00 (18.00, 23.00) points, respectively. Most of them were treated with neoadjuvant chemotherapy combined with gastrectomy (81.63%), where the chemotherapy was mainly based on paclitaxel (75.00%). Around half of the subjects underwent subtotal gastrectomy (51.02%). After discharge, 57.14% patients took oral nutrition supplements (ONS) during the follow up (Tables 1 and 2)

Body weight change before and during three months after gastrectomy

The median body weight loss in GC patient at T1, T2, and T3 was 9.85% (7.41%, 11.76%), 12.14% (8.70%, 12.14%), and 11.63% (8.51%, 14.40%), respectively, for the whole cohort (Fig. 1). The GEE results showed a significant effect of time (Wald χ2 = 333.917, P < 0.001), indicating a decrease in body weight over time. According to the GEE parameter estimation, patient’s body weight decreased by 9.2% at T1 (Wald χ2 = 271.173, P < 0.001), 11.0% at T2 (Wald χ2 = 277.267, P < 0.001), and 11.4% at T3 compared to baseline at T0 (Wald χ2 = 284.076, P < 0.001) (Table 3).

Factors contributed to body weight change during three months after gastrectomy

Each study variable entered GEE separately for univariate analysis. The results of the GEEs for univariate analysis showed that total gastrectomy patients had greater body weight loss compared with distal gastrectomy patients (β = 3.1%, P = 0.003), patients with postoperative complication was associated with greater body weight loss than patients without postoperative complication (β = 3.0%, P = 0.010), BMI ≥ 25 kg/m2 was associated with greater body weight loss compared with BMI < 18.5 kg/m2 (β = 3.9%, P = 0.080), and higher nutrition practice score was associated with greater weight loss (β = 0.3%, P = 0.070) (Table 4). Therefore, those variables with P value less than 0.1 (operation type, preoperative BMI, post-operative complication, and nutrition practice score) were entered into GEE to do multivariable analysis. The results of GEE multivariable analysis indicated that surgery type and preoperative BMI were contributing factors of body weight change, while other variables, including postoperative complication and nutrition practice were not relevant. Compared with distal gastrectomy patients, total gastrectomy patients experienced greater body weight loss (β = 2.8%, P = 0.014). Compared with patient with BMI<18.5 kg/m2, patients with BMI ≥ 25 kg/m2 experienced greater body weight loss (β = 4.5% P = 0.026) (Table 5).

Table 1 Demographic, clinicopathology, treatment characteristics of patients with GC.

Variables	N (percentage)	
Age		
<65 y	37 (75.51%)	
≥65 y	12 (24.49%)	
Gender		
Male	22 (44.90%)	
Female	27 (55.10%)	
Marriage status		
Single	1 (2.04%)	
Married	45 (91.84%)	
Devoiced or widowed	3 (6.12%)	
Education level		
Elementary or below	9 (18.37%)	
Middle	8 (16.33%)	
High	11 (22.45%)	
College or above	21 (42.86%)	
Religion		
None	42 (85.71%)	
Buddhism	7 (14.29%)	
Preoperative BMI		
Underweight	5 (10.20%)	
Normal weight	21 (42.86%)	
Overweight	10 (20.41%)	
Obesity	13 (26.53%)	
Cancer stage		
II	2 (4.08%)	
III	28 (57.14%)	
IV	19 (38.78%)	
Cancer therapy		
Surgery	1 (2.04%)	
Surgery + Chemotherapy	8 (16.33%)	
Neoadjuvant chemotherapy +Surgery	40 (81.63%)	
Type of operation		
Distal	25 (51.02%)	
Total	24 (48.98%)	
Type of chemotherapy		
Base on paclitaxel	37 (75.51%)	
Base on xeloda	1 (2.04%)	
Base on oxaliplatin	4 (8.16%)	
Base on irinotecan	1 (2.04%)	
Base on S-1	6 (12.24%)	
ONS		
Without	28 (57.14%)	
With	21 (42.86%)	
Complication		
Without	36 (73.47%)	
With	13 (26.53%)	
Notes.

BMI body mass index

ONS oral nutritional supplements

Discussion

Body weight change after gastrectomy

GC Patients often experience body weight loss after gastrectomy, which involves complicated mechanism and a lot of factors, including remnant stomach, malabsorption, hormonal changes, and life style (Aoyama, 2019). Some scholars deemed that subtotal or total excision of gastric tissue negatively influence digestion and absorption of food inevitably. Also, operation itself may result in anorexia, thereby leading to body weight loss (Kim et al., 2019). Eom et al.( 2018) found that postoperative body weight loss and anemia might originate from altered absorptive function and metabolic change after gastrectomy rather than decreased nutrient intake. Aoyama et al. (2016) revealed that mean body weight loss at 1 month reached 3.4 kg (5.90% of body weight) compared with preoperative mean body weight in Japanese subjects . The team of Aoyama et al. also found that mean body weight loss value at 1 month and 3 months after surgery were 6.5% and 9.0% respectively in the elderly, while 6.0% and 8.1% respectively in the non-elderly from the preoperative value, but the difference of weight loss over time between elderly and non-elderly was not significant (Aoyama et al., 2019). In China, researchers found that mean body weight loss up to 5.8–8.8 kg at 1 month (Shuai et al., 2019), and 6.33 kg at 3 months (Tang et al., 2018) after gastrectomy compared with preoperative body weight. Our results show that the percentage of body weight loss up reached 9.2% (5.67 kg) at 1 month, 11.0% (6.83 kg) at 2 months, and 11.4% (7.14 kg) at 3 months after surgery from preoperative mean body weight. The higher body weight loss at three months in our study is possibly related the fact that most of the enrolled patients were advanced staged and underwent extensive chemotherapy.

Table 2 CCI and KAP characteristics of patients with GC.

Variables		Median	25% quartiles, 75% quartiles	
CCI		5.00	3.00, 6.00	
KAP	Knowledge	12.00	8.00, 13.00	
	Attitude	15.00	14.00, 18.00	
	Practice	21.00	18.00, 23.00	
Notes.

CCI Charlson comorbidity index

KAP Nutrition Knowledge, Attitude and Practice

Figure 1 The loss of body weight after gastrectomy at T1, T2, and T3 in this study.

T0: before surgery, T1: 1 month after gastrectomy, T2: 2 months after gastrectomy, T3: 3 months after gastrectomy, Bars: Median, boxes: lower and upper quartiles, lines: minimum and maximum.

Table 3 Percentage of body weight change over time before and during 3 months after gastrectomy in GC patients.

Variables	β	SE	95% CI	Wald χ2	P	
			lower	upper			
Time					333.917	<0.001	
T3	11.4%	6.8%	10.1%	12.8%	284.076	<0.001	
T2	11.0%	6.6%	9.7%	12.3%	277.267	<0.001	
T1	9.2%	5.6%	8.1%	10.3%	271.173	<0.001	
T0	—						
Notes.

T0 before surgery

T1 1 month after gastrectomy

T2 2 months after gastrectomy

T3 3 months after gastrectomy

BMI body mass index

β partial regression coefficient

SE standard error

CI confidence interval

Table 4 Association of study variables with body weight change three months after gastrectomy in univariate analysis.

Variables	β	SE	95%CI	Wald χ2	P	
			lower	upper			
Age							
<65 y	1.4%	1.4%	–1.3%	4.2%	1.071	0.301	
≥65 y	—						
Gender							
Female	<0.1%	1.1%	–2.2%	2.3%	0.001	0.977	
Male	—						
Marriage status							
Single, or devoiced or widowed	–1.7%	2.5%	–6.5%	3.1%	0.472	0.492	
Married	—						
Education level					2.275	0.321	
Middle or below	1.9%	12.8%	–0.6%	4.4%	2.197	0.138	
High	1.4%	1.5%	–1.5%	4.3%	0.847	0.358	
College or above	—						
Religion							
Buddhism	2.0%	1.6%	–1.0%	5.0%	1.645	0.200	
No religion	—						
Cancer stage					1.514	0.469	
IV	3.3%	3.1%	–2.7%	9.4%	1.158	0.282	
III	2.2%	3.0%	–3.6%	8.1%	0.550	0.458	
II	—						
Type of operation							
Total	3.1%	1.1%	1.0%	5.2%	8.551	0.003	
Distal	—						
Treatment type					0.585	0.747	
Surgery	0.4%	0.7%	–0.8%	1.7%	0.442	0.506	
Surgery + Chemotherapy	–0.1%	1.7%	–3.4%	3.1%	0.008	0.929	
Neoadjuvant chemotherapy + Surgery	—						
ONS							
With	0.9%	1.1%	–1.3%	3.2%	0.667	0.414	
Without	—						
Preoperative BMI					8.993	0.029	
Obesity	3.9%	2.2%	–0.5%	8.2%	3.063	0.080	
Overweight	0.8%	2.5%	–4.1%	–5.7%	0.092	0.761	
Normal weight	1.6%	2.3%	–3.0%	–6.1%	0.456	0.499	
Underweight	—						
Complication							
With	3.0%	1.1%	0.7%	5.2%	6.700	0.010	
Without	—						
CCI	0.2%	0.2%	–0.2%	0.6%	0.878	0.349	
Knowledge	0.1%	0.1%	–0.2%	0.3%	0.260	0.610	
Attitude	0.3%	0.2%	–0.1%	0.6%	2.012	0.156	
Practice	0.3%	0.2%	<0.1%	0.6%	3.288	0.070	
Notes.

ONS oral nutritional supplements

BMI body mass index

CCI Charlson comorbidity index

β partial regression coefficient

SE standard error

CI confidence interval

Table 5 Association of study variables with body weight change three months after gastrectomy in multivariable analysis.

Variables	β	SE	95% CI	Wald χ2	P	
			lower	upper		
Intercept	11.2%	2.9%	5.5%	7.0%	14.624	<0.001	
Time					19.780	<0.001	
T1	–0.2%	0. 5%	–3.2%	–1.2%	19.438	<0.001	
T2	–0.4%	0. 3%	–1.0%	0.2%	1.999	0.157	
T3	—						
Preoperative BMI					12.662	0.005	
Obesity	4.5%	2.1%	0. 5%	8.4%	4.928	0.026	
Overweight	1.9%	2.3%	–2.6%	6.3%	0.673	0.412	
Normal weight	1.3%	2.0%	–2.6%	5.3%	0.430	0.512	
Underweight	—						
Type of operation							
Total	2.8%	1.2%	0.6%	5.1%	6.027	0.014	
Distal	—						
Complication							
With	1.0%	1.3%	–1.6%	3.6%	0.533	0.465	
Without	—						
Practice	0.2%	0.1%	<0.1%	0.5%	3.010	0.083	
Notes.

T1 1 month after gastrectomy

T2 2 months after gastrectomy

T3 3 months after gastrectomy

BMI body mass index

β partial regression coefficient

SE standard error

CI confidence interval

The variation of postoperative body weight loss in different studies indicated that postoperative body weight was affected by a lot of factors. Aoyama et al. (2016) found that body weight loss during the first week after surgery was significantly greater than that during subsequent 3 weeks, and loss of lean body mass accounted for a significant part of body weight loss during the first week due to surgical stress . Abdiev et al. (2011) discovered that both fat and muscle mass reduction accounted for body weight loss during the first month after gastrectomy, and fat mass loss followed was responsible for weight loss after that period . Therefore, since energy deficiency and fat mass consumption rather than protein consumption are more prominent after discharge, continuous oral intake and exercise intervention should be given to GC patients after discharge to ensure the sufficient energy intake. In addition, body weight loss at 2 months was not significantly different from at 3 months after gastrectomy in this study, indicating that clinician should pay attention to the underlying causes of weight loss and direct nutrition intervention before 2 months after gastrectomy as soon as possible.

Factors contributed to body weight change after gastrectomy

The results of GEE showed significant main effect of preoperative BMI categories on body weight change after gastrectomy. Same as the findings in previous study (Aoyama et al., 2019; Shuai et al., 2019; (Tang et al., 2018)), we found that the percentage of body weight loss showed a downward trend as a whole within 3 months after gastrectomy, and the percentage of body weight loss within the first month postoperative period was significantly more than that in the third month postoperative. Similar to the findings in the study of Tanabe et al. who found that higher preoperative BMI, was independent predictors of greater body weight loss at the first one year mark after gastrectomy (Tanabe et al., 2017). We found that BMI ≥ 25 kg/m2 was risk factors to body weight loss during three months after gastrectomy in GC patients. However, the result was contract with the study by Park et al. (2018) who found that preoperative BMI < 23 kg/m 2were associated with severe weight loss after gastrectomy . Similar to previous studies (Davis et al., 2016; Nishigori et al., 2017; Segami et al., 2018), we found that total gastrectomy was a risk factor for postoperative weight loss. Previous study observed that decreased calorie intake after total gastrectomy compared with distal gastrectomy (Noguchi et al., 1992), which may explain total gastrectomy patients experienced more body weight loss.

Similar to the findings in the study of Davis et al. (2016), we found that postoperative complication was not found to be independent factor on postoperative body weight loss in this study. While Segami et al. found that surgical complication was a significant risk factor for severe body weight loss the first month after gastrectomy (Segami et al., 2018). The conflict results prompt the complexity of weight loss in GC patients after gastrectomy. As for higher nutrition knowledge, attitude, and practice score were positively related with higher self-efficacy (Jing, Wei-li & Xin-qiong, 2016), which is a vital theory to implement nutrition education, less weight loss was expected to appear in patients with higher nutrition knowledge, attitude, and practice score in this study. However, there was no significant effect of nutrition knowledge, attitude, and practice on postoperative body weight loss. Consistent with some scholars’ findings (Aoyama et al., 2019; Davis et al., 2016), our study verified that age was not a risk factor for postoperative body weight loss during 3 months after gastrectomy. Recently, Aoyama et al. found that postoperative lean body mass decreased significantly in elderly compared with non-elderly rather than body weight (Aoyama et al., 2019).

There were some limitations in this study. Firstly, this was a single center study, 95.92% patients involved in our study were with cancer stage III or IV GC. So, the results that total gastrectomy and preoperative BMI were factors contributing to body weight change may not generalizable to patients with stage I or II. Secondly, we did not collect data about patient nutritional intake amount during 3 months after gastrectomy. So, the nutrition therapy factors on body weight change may be ignored in our study. Thirdly, the body weight at each point was reported by the patient self.

Conclusion

GC patients experienced significant body weight loss during three months after gastrectomy. Total gastrectomy and BMI ≥ 25 kg/m2 were risk factors to postoperative body weight loss for GC patients in this study. Given adverse effect of weight loss after surgery on overall survival and quality of life, the results suggested hinted that clinician should pay attention to postoperative nutrition status of GC patient undergoing total gastrectomy and obesity patients.

Supplemental Information

Supplemental Information 1 Original data

Click here for additional data file.

Additional Information and Declarations

Competing Interests

Author Contributions

Human Ethics

Data Availability

The authors declare there are no competing interests.

Qiuju Tian conceived and designed the experiments, performed the experiments, analyzed the data, prepared figures and/or tables, authored or reviewed drafts of the paper, and approved the final draft.

Liyuan Qin performed the experiments, analyzed the data, prepared figures and/or tables, and approved the final draft.

Weiyi Zhu analyzed the data, authored or reviewed drafts of the paper, and approved the final draft.

Shaojie Xiong performed the experiments, authored or reviewed drafts of the paper, and approved the final draft.

Beiwen Wu conceived and designed the experiments, performed the experiments, authored or reviewed drafts of the paper, and approved the final draft.

The following information was supplied relating to ethical approvals (i.e., approving body and any reference numbers):

Ethics approval of this study was obtained from Ruijin Hospital Ethics Committee, Shanghai Jiao Tong University School of Medicine, China.

The following information was supplied regarding data availability:

Datais available at Figshare:

Tian, Qiuju; Qin, Liyuan; Wu, Beiwen; Zhu, Weiyi; Xiong, Shaojie (2019): body weight change dataset. figshare. Dataset. https://doi.org/10.6084/m9.figshare.9918020.v1.

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
