# Peer review of "Analysis of factors contributing to postoperative body weight change in patients with gastric cancer: based on generalized estimation equation"

_PeerJ, doi:10.7717/peerj.9390_

## Round 0.1 · original submission · Major Revisions

This manuscript seems to have had highly divergent reviews across the four reviewers, so we feel it is appropriate to give you the opportunity to respond to these comments. When making your point by point revisions, please make sure you address all those of the reviewers, in particular reviewer number one, who was pinpointed a considerable number of areas in which the manuscript can be improved.

Reviewer 1 ·

Basic reporting

There are many basic reporting problems with this article:

1) The abbreviation TNM is used extensively but never explained.

2) In the abstract, the P values of body weight at T1 and T2 have an unusual symbol with Greek letters instead of <.

3) The abstract is clumsily written. The sentences "The result of generalized estimation equation showed that female, Buddhism religion, and lower body mass index significantly influenced postoperative body weight changes independently (all P < 0.05). Specifically, compared with male patients, female patients experienced more weight loss. Compared with Buddhism religion, the non-region were protective factors for postoperative body weight change. Taking obesity patients as reference, underweight patients experienced the largest body weight loss, followed by normal weight patients, and overweight patients" are largely redundant and should be summarised in one sentence i.e. "being female, of Buddhism religion, and having lower body mass index were independent predictors of greater postoperative weight loss."

4) English language writing is poor throughout and requires significant revision by a native English speaker. Particularly problematic terms include "dig out", "zoomed in on", "dug out that", "come to", "comes to"- none of these are acceptable in formal scientific writing.

5) Furthermore, the phrase "no less than 10%" is confusing and not used conventionally in scientific writing. Why not just use the phrase "10% or greater?"

6) Similarly, the phrase "surgical complication grade no less than two" is confusing and not used conventionally in scientific writing. Why not just use the phrase "surgical complication grade of 2 or greater?" Also, in this citation, surgical complication grade is not explained- which validated scale is being used exactly to determine surgical complications?

7) The sentence "After surgical treatment, appetite and food intake in GC patients changed, resulting in the recovery of nutritional status last up to one year(Ryu & Kim 2010), where body weight loss usually accompanied with" is particularly poorly written and requires revision.

8) In the first 2 sentences of the "Results" paragraph, there is inconsistent use of Roman numerals and words for numbers. "six" should be "6" and "one" should be "1"

9) The sentence "More than half of them were female (55.10%), married (91.80%), without religious brief (85.7%), and with a stage III cancer (57.1%)" is poorly phrased. It implies that more than half the patients met all the above criteria.

10) The sentences "specifically, the body weight continued to reduce 1.16 kg from T1 to T2, and 0.31kg from T2 to T3" are redundant.

11) The three sentences "Lower preoperative BMI was a risk factor for body weight loss during three months after gastrectomy in GC patients. The lower the preoperative BMI was, the higher the body weight loss was. Patients with underweight normal weight, and overweight experienced significantly body weight loss compared with obese patients" are all saying the same thing. Two of these sentences should be deleted for redundancy.

12) Table 4 has missing information in the legend, including explaining T1/T2/T3 and II/III/IV, and also the meaning of "B" and "SE".

13) Table 3 has no legend, including explaining T1/T2/T3 and also the meaning of "B" and "SE".

Experimental design

METHODOLOGICAL PROBLEMS
1) The abstract states that covariates were entered into generalized estimation equation after univariate analysis. However in the main body of the text, univariate analysis is never described. Instead, the authors state, confusingly, that generalized estimation equations are used for multivariate analysis.

2) The sentence "And time variable on postoperative body weight change should be considered in the model" is poorly written as the authors have not referred to any particular mathematical models by this point in the manuscript. Furthermore, the authors actually fail to discuss any further time variables on postoperative body weight change in the results or the conclusion, even though Table 4 shows that a lower weight at T1 is an independent predictor of greater weight loss at T3.

3) It is unclear why, according to the statistical analysis, data is presented as median combined with minimum and maximum. This is not standard practice in scientific writing: why do the authors not use median, quartiles and interquartile range?

4) Throughout the manuscript and tables/figures, body weight is referred to as "body weight", "average body weight", etc. It should always be consistently referred to as mean body weight.

5) The authors should not use mean absolute changes in body weight or BMI as variable or outcomes, as this may be dependent on patient baseline weight. They should use mean percentage changes in body weight instead, just like most of the previous studies that they cite (e.g. Climent et al, Aoyama et al.

6) The choice of generalized estimation equations as a statistical method is extremely difficult to comprehend in this study, and poorly explained.

The sentence "QIC value in independent form was smaller than that in the other four forms in this study" is confusing- what are the "other four forms" that the authors refer to?

The entire explanation of the GEEs in the paragraph entitled "Factors contributed to body weight change during three months after gastrectomy" is confusing. Firstly the authors state that the results of GEEs showed that sex, religion, TNM, CCI significantly influenced body weight changes. However then these variables are entered into GEEs for mutlivariable analysis- this is confusing- were two sets of GEEs involved? Furthermore, if only sex, religion, TNM, CCI were entered into GEE to do multivariate analysis, why does the analysis then include other variables such as preoperative BMI, ONS, nutrition knowledge/attitude/practice?

Furthermore, the copula function of GEE is not explained at all. Is this a mathematical model to predict absolute weight at 3 months? Or absolute BMI at 3 months? At what time point are the BMI in the copula function measured? And what does the variable "1 month after gastrectomy" in the copula function mean? In what practical way can clinicians use this copula function?

Why was the crucial variable of partial versus complete gastrectomy not analysed in the multivariate analysis?

I propose that for such a small, simple study, with less than 50 patients and only three time points, the authors abandon such a complex statistical methodology which introduces significant confusion. Instead, they should simply perform univariate linear regression with patient variables (including baseline variables, as well as weight change at T1 and T2) and a continuous outcome of weight loss at T3. After this, multivariate linear regression of variables where P <0.1 should then be performed to identify independent predictors of weight loss at T3.

Validity of the findings

DISCUSSION/CONCLUSION PROBLEMS
1) The authors do not provide an adequate explanation of why Buddhist religion may lead to greater weight loss. The statement "Buddhist refuses or limits the meat intake...aggravating nutritional status and body weight loss" is an unproven generalisation, not supported by cited studies or data in this study actually assessing meat intake. It is therefore speculation and should be deleted.

2) The sentence "Actually, the compliance of ONS is poor in cancer patients in clinical practice, which may explain why ONS in this study did not showed significant effect on reducing body weight loss" is unproven, not supported by cited studies, and of course not supported by data in this study as the authors acknowledge their failure to record ONS intake quantity. It is therefore speculation and should be deleted.

3) The discussion of limitations is poor. The authors fail to acknowledge in this section that they did not collect data about patient nutritional intake, let alone ONS intake, post surgery. The statement "this was a single center study, most of patients were with advanced GC admitted to our hospital, which may come to patient bias" is poorly explained. What "patient bias" are they referring to? Is it selection bias- and if so, does it affect the internal validity or external validity of the study, or both?

Finally, the statement "we did not analyze the relationship between body weight change and adverse event, as we just followed three months" is misleading. The authors failed to collect any information of adverse events after the surgery during the hospital recovery period, which would have been easily available from patient records and could have been an important variable predicting post-operative weight loss, even confounding all of their conclusions (as they themselves acknowledge in the previous sentence quoting Zhao et al. 2018).

Reviewer 2 ·

Basic reporting

1) Some errors in editing and grammar were found. I mentioned these in the "General comments for the author".
2) References well support the plot.
3) Well structured and raw data were also shared.
4) Relevant results.

Experimental design

1) Within aims and scope of the journal
2) Relevant and meaningful results.
3) Proper investigations for clinical questions.
4) There is room for improvement in M & M section and I mentioned in the "General comments for the author".

Validity of the findings

Relevant.

Additional comments

Overall, this is an interesting study describing factors associated with postoperative weight loss in gastric cancer patients. Although the conclusion of the study should be validated in a larger number of cohorts in the future, this manuscript is suitable for publication. There are several minor suggestions to improve the quality of the paper.

Materials and methods
1) There is no specific description of how the body weight was measured (e.g. self-report and so on). Please make up for it. If weight measurements were not performed uniformly, please indicate in the discussion section as a limitation of the study.

Result
1) Are there any cases of relapse? Since recurrence of cancer can contribute to weight loss, please reanalyze the effects of recurrence on weight loss if any cases exist. If no recurrence was observed during follow-up period, please indicate this fact.
2) The first section of the Result has no title. Please add the title such as “baseline characteristics of the study population”.
3) Mark the expansion for PTX if this word was first used in the manuscript.
4) In the 2nd section of Result, there should be a gap between the number and the unit (e.g. kg).
5) In Table 1, upper case should be used for ‘f’emale and so on.
6) In Table 1, what is the ‘type of C’?
7) In Table 3 and 4, please indicate the expansion for ‘B’, ‘SE’, and ‘CI’?

Discussion
1) The effect of religion on postoperative weight loss cannot be generalized easily. Please include this as a limitation of the study.

Reviewer 3 ·

Basic reporting

No comment

Experimental design

Not clear description of this study's design: prospective cohort or retrospective electric medical record review.

Small number of patients - not enough to conclude and sub-analyze

Very short follow-up period

Validity of the findings

No novelty - the fact of weight loss after gastrectomy is well known

Reviewer 4 ·

Basic reporting

no comment

Experimental design

no comment

Validity of the findings

no comment

Additional comments

The authors investigated factors contributing to postoperative body weight change in gastric cancer patients, and showed female, Buddhism religion, and lower preoperative BMI were risk factors to post-op body weight loss by using generalized estimating equation analysis. As they mentioned in the introduction section, it has been well known that body weight loss after surgery was associated with an unfavorable long-term outcome in gastric cancer patients. That's why it would be useful if surgeons predicted such patients, and commenced exercise and nutrition intervention preoperatively. At a glance, this is a well-written paper that presents interesting data, however, I have the following concerns.
1) In clinical practice, after TOTAL GASTRECTOMY, the severe body weight loss is commonly observed. But, the authors didn’t present the data and their interpretation about patients undergoing total gastrectomy. I suggest that the authors should touch on the issue and details in the manuscript.
2) In the discussion section, they interpreted the risk factor to weight loss of Buddhism religion as the limitation of meal intake. Their interpretation of that is not plausible. It would be useful if the authors gave Buddhist patients’ eating habits or nutritional knowledge, attitude and practice.

---

## Round 0.2 · Major Revisions

While two reviewers are happy with your corrections, the first reviewer still has a number of concerns that you need to address.

Reviewer 1 ·

Basic reporting

Generally improved since last review. Small errors in English grammar will likely be cleaned up in the editing process.

Some important vocabulary changes needed in the discussion though:

1) line 268- "considered to" should be "expected to"

2) line 277- "Coherent" should be "consistent"

3) line 284- "suitable for" should be "generalizable to".

4) in the tables "stand error" should be "standard error"

Experimental design

Methodology has been better explained now.

1) However in my previous review, I wrote that "The authors should not use mean absolute changes in body weight or BMI as variable or outcomes, as this may be dependent on patient baseline weight. They should use mean percentage changes in body weight instead, just like most of the previous studies that they cite (e.g. Climent et al, Aoyama et al.) This has not yet been addressed by the authors.

Studies such as Climent et al, Aoyama et al, Davis et al, Kubo et al all express weight change for each given patient as a % weight change compared to baseline.

This is a much more meaningful measurement than absolute weight change in kilograms, which is what the authors of the current study use. It puts the absolute weight change in the context of the patient's baseline weight. For example, a 100kg person losing 5kg is clinically of far less concern than than a 40kg person losing 5kg, even it is seems that their absolute weight loss is identical. The first person has lost 5% of their body weight. The second person has lost 12.5% of their body weight. I understand that there is not so dramatic a range of weights in the patients in the current study. However, the mathematical principle remains the same: using % weight change is a more appropriate scientific measurement, and has extensive precedent in the worldwide literature quoted by the authors. It is unclear why the authors have chosen to persist with expressing measurements with absolute weight change in kilograms.

The authors do find that a lower baseline weight is independently associated with greater absolute weight loss in kg. I argue that performing their analysis according to % body weight loss would actually accentuate and statistically strengthen their findings! It would show that the % weight loss in the underweight group is far higher than the % weight loss in the normal weight or overweight group, and the difference is of much greater magnitude than when measuring absolute weight loss in kg. It would make this an even more convincing and powerful article. If the authors choose not to take up my suggestion, they should at least include this as a major methodological limitation.





2) The explanation of GEEs in univariate analysis from line 173 to 180 needs to be in the Statistics section of Materials and Methods, not in the Results. Furthermore, it needs to include explicit mention of each study variable that was studied in univariate analysis. Furthermore, it needs to mention that multivariate analysis was then performed, which variables from the univariate analysis were chosen to undergo multivariate analysis, and on what basis they were chosen. That is scientific convention in the statistics section of an article.

3) These sentences from line 180 are confusing "The results of the GEEs for univariate analysis showed that sex, religion, TNM, CCI significantly influenced body weight changes (all P < 0.05), and education level, operation type (partial vs completely gastrectomy), chemotherapy type, cancer therapy type, and perioperative complication showed no statistically significance with the body weight change. Therefore, those variables (sex, religion, TNM, CCI) were entered into GEE to do multivariable analysis, and the results of GEE multivariable analysis indicated that sex, religion, and preoperative BMI were contributing factors of body weight change (P < 0.05), while other variables, such as TNM stage,ONS, CCI, patients' nutrition knowledge, nutrition attitude, and nutrition practice were not relevant." It suggests that many variables were included in the multivariable analysis that were not included in the preceding univariate analysis such as preoperative BMI, ONS, patients' nutrition knowledge, nutrition attitude, and nutrition practice. In fact, the authors confirm this in the rebuttal letter to me. This is not, to my knowledge, the correct way to perform univariate and multivariate analysis. Univariate analysis should start by including all the possible study variables. Then, only those variables that meet a pre-specified level of statistical significance on univariate analysis go on to be included in multivariate analysis. Why did the authors not do this?

4) Continuing from point 3 above, there are demographic variables that were collected but not included in analysis such as marriage status, smoking, and drinking. Firstly, how exactly was 'smoking' and 'drinking' defined? Does 'drinking' mean alcohol? Secondly, what was the point of collecting or presenting these variables if they were not included in the analysis? I believe they should also be included in the univariate +/- multivariate analysis (as long as 'smoking' and 'drinking' can be adequately defined).

5) Continuing from point 3 above, table 4 is incomplete and confusing. It states p values but does not state whether this is on univariate or multivariate analysis. Despite it being called "Parameters associated with body weight change", it presents many variables that actually were not associated with body weight change as they did not reach statistical significance. If, indeed, the authors instead wanted to provide a table that included all variables, they should change the title, and also include many variables that were omitted including education level, operation type (partial vs completely gastrectomy), chemotherapy type, cancer therapy type, and perioperative complication.

I suggest the authors update table 4 so it includes every single study variable, one column for p values of univariate analysis, and one column for p values of multivariate analysis. This could be titled "Association of study variables with body weight change three months after gastrectomy". It should be arranged intuitively: with variables that reached significance on multivariate analysis clustered together in rows at the top of the table, and variables that did not reach significance clustered together in rows separately at the bottom of the table.

Validity of the findings

Nothing to add here

Reviewer 2 ·

Basic reporting

1) Some errors in editing and grammar were found. I mentioned these in the "General comments for the author".
2) References well support the plot.
3) Well structured and raw data were also shared.
4) Relevant results.

Experimental design

1) Within aims and scope of the journal
2) Relevant and meaningful results.
3) Proper investigations for clinical questions.
4) There is room for improvement in M & M section and I mentioned in the "General comments for the author".

Validity of the findings

Relevant.

Additional comments

All issues raised by me have been satisfactory addressed.

Reviewer 4 ·

Basic reporting

no comment

Experimental design

no comment

Validity of the findings

no comment

Additional comments

no comment

---

## Round 0.3 · Minor Revisions

While you have attended to many of the comments of the reviewers, you have not really attended sufficiently to the constructive criticisms of this reviewer. I, therefore, suggest you strongly consider taking on board the advice of this reviewer if you wish this paper to be considered for publication in this journal.

Reviewer 1 ·

Basic reporting

There are confusing contradictory statements in the discussion: line 238, "there was no significant difference in weight loss between total and subtotal gastrectomy in this research. The result was similar to previous studies, where total gastrectomy was a risk factor for postoperative weight loss." These two sentences are directly contradictory. Also the first sentence is wrong. There was a significant difference in weight loss between total and subtotal gastrectomy in this research, as reported all throughout the manuscript.

Experimental design

My previous comments asked "how exactly was 'smoking' and 'drinking' defined?"

The answer from the authors has been inadequate. I quote: "Smoking was defined as with the habits of smoking. Drinking was defined as with the habit of alcohol drinking."

This is inadequate. What does 'with the habit of smoking mean'? Does it mean the patients are current smokers? Or current or former smokers? Did the authors apply a quantitative number of pack-years of smoking? If the authors cannot show they have made any attempt to quantify the amount of smoking exposure, then smoking should be removed from the manuscript altogether.

What does "with the habit of alcohol drinking mean'? Did the authors apply a quantitative number of alcohol intake e.g in mean grams of ethanol per day? Or in mean standard drinks per day? If the authors cannot show they have made any attempt to quantify alcohol intake, then alcohol should be removed from the manuscript altogether.

Validity of the findings

The reporting of the variables associated with weight loss in univariate and multivariate analysis in lines 180 to 190 needs to be clearer in terms of the direction of effect.

The authors mention operation type, preoperative BMI, postoperative complication and nutrition practice score as influencing body weight change. For each variable, what was the direction of effect? Did total gastrectomy or partial gastrectomy associate with greater weight loss? Did higher pre-operative BMI or lower pre-operative BMI associate with greater weight loss? Did more or less complications associate with greater weight loss? Did higher or lower nutrition practice score associate with greater weight loss?

Similarly, in the multivariate analysis, total gastrectomy and preoperative BMI were contributing factors of body weight change. What was the direction of effect? Did total gastrectomy or partial gastrectomy associate with greater weight loss? Did higher pre-operative BMI or lower pre-operative BMI associate with greater weight loss?

---

## Round 0.4 · accepted · Accept

I'd like to thank the reviewers for their hard work in making the required amendments to this manuscript, that is now acceptable for publication in PeerJ.

Reviewer 1 ·

Basic reporting

No further issues.

Experimental design

No further issues. All my criticisms answered

Validity of the findings

No further issues. All my criticisms answered